# Q-Learning Based Joint Energy-Spectral Efficiency Optimization in Multi-Hop Device-to-Device Communication

**DOI:** 10.3390/s20226692

**Published:** 2020-11-23

**Authors:** Muhidul Islam Khan, Luca Reggiani, Muhammad Mahtab Alam, Yannick Le Moullec, Navuday Sharma, Elias Yaacoub, Maurizio Magarini

**Affiliations:** 1Thomas Johann Seebeck Department of Electronics, School of Information Technology, Tallinn University of Technology, Ehitajate tee 5, 19086 Tallinn, Estonia; muhammad.alam@ttu.ee (M.M.A.); yannick.lemoullec@ttu.ee (Y.L.M.); navuday.sharma@taltech.ee (N.S.); 2Dipartimento di Electtronica e Informazione, Politecnico di Milano, Via Ponzio 34/5, 20133 Milano, Italy; luca.reggiani@polimi.it (L.R.); maurizio.magarini@polimi.it (M.M.); 3Faculty of Computer Studies, Arab Open University, Beirut 2058 4518, Lebanon; eliasy@ieee.org

**Keywords:** joint energy-spectral efficiency (ESE), device-to-device (D2D), public safety networks, pervasive public safety communication, Internet of Things (IoT)

## Abstract

In scenarios, like critical public safety communication networks, On-Scene Available (OSA) user equipment (UE) may be only partially connected with the network infrastructure, e.g., due to physical damages or on-purpose deactivation by the authorities. In this work, we consider multi-hop Device-to-Device (D2D) communication in a hybrid infrastructure where OSA UEs connect to each other in a seamless manner in order to disseminate critical information to a deployed command center. The challenge that we address is to simultaneously keep the OSA UEs alive as long as possible and send the critical information to a final destination (e.g., a command center) as rapidly as possible, while considering the heterogeneous characteristics of the OSA UEs. We propose a dynamic adaptation approach based on machine learning to improve a joint energy-spectral efficiency (ESE). We apply a Q-learning scheme in a hybrid fashion (partially distributed and centralized) in learner agents (distributed OSA UEs) and scheduler agents (remote radio heads or RRHs), for which the next hop selection and RRH selection algorithms are proposed. Our simulation results show that the proposed dynamic adaptation approach outperforms the baseline system by approximately 67% in terms of joint energy-spectral efficiency, wherein the energy efficiency of the OSA UEs benefit from a gain of approximately 30%. Finally, the results show also that our proposed framework with C-RAN reduces latency by approximately 50% w.r.t. the baseline.

## 1. Introduction

In the context of the cellular networks, from Advanced Long Term Evolution (LTE-A) to fifth-generation of mobile communication (5G), the increasing number of devices keeps pushing the demand for higher spectral and energy efficiencies. In this direction, 5G provides a roadmap for increased resource efficiency and energy efficiency, greater reliability, and low latency solutions [1]. In particular, Device-to-Device (D2D) communication is regarded as a key technology in 5G wireless systems for providing services that include live data and video sharing [2]. D2D communication allows user devices (UEs) that are in close proximity to exchange information over a direct link, which can be operated as an underlay to LTE-A networks by reusing the spectrum resources. D2D communication is not only useful for local communication but also for extending the range of a base-station (BS) to out-of-coverage UEs. This opportunity can be provided by D2D-based relays. Relay UEs help to communicate with the BS and other out-of-coverage UEs, as standardized in 3GPP Release 13 [3]. Two key features of D2D proximity service (ProSe) are [4]: (i) UEs in close physical proximity are able to discover the existence of each other through network assisted discovery and (ii) direct communications between two UEs, with or without inclusion of the control from the network, can be enabled using a direct interface called Sidelink.

In this work, we exploit these concepts in a scenario where UEs are partially connected with the network infrastructure, damaged or deactivated, and there is the necessity of conveying the connectivity to a given destination. This is a crucial case in public safety networks (PSN), which are mission critical wireless networks for emergency scenarios [5] that create a link between the persons in the area, including the rescue teams, and a national command center for sharing mission-critical information. However, traditional PSNs are not designed to cope with cases during which the UEs are partially connected with the network infrastructure, e.g., due to on-purpose deactivation from the authorities (e.g., in case of terrorist activity) or physical damage. In this context, D2D communication is seen as a solution for extending the coverage of the sites that remain active in such partial coverage scenarios. Therefore, our proposed setup considers multi-hop D2D communication in the reference context of PSNs, e.g., for disasters or terrorist attacks where the cellular BS (i.e., eNodeB or gNodeB) becomes non-functional or fully destroyed. We also observe that, in such a situation, also Unmanned Aerial Vehicles (UAVs) could be deployed in order to assist On-Scene Available (OSA) UEs in disseminating the information, such as user ID, location, and images, to the command center without going through the BSs.

Several challenges regard the end-to-end network connectivity, which should be guaranteed till to the external command center where all the critical information are collected:In a traditional scenario, the BS would allocate the resources to the cellular and D2D users; dedicated resource allocation and proper power allocation would be applied at the D2D devices and interference and reliability would be controlled. However, in the underlay mode (as in our scenario), resource allocation and interference management pose a challenge due to the absence of BS control, meaning that these tasks have to be executed on the OSA UEs and UAVs if present (the latter act as RRHs).Reliable connectivity and routing are not generally well suited to severe link conditions, variable transmission range, heterogeneous resources and unique mobility patterns like those in a disaster scenario.Multi-hop communication is essential for such a scenario in order to disseminate the critical information. Finding the best routing path is challenging due to harsh propagation conditions and severe links obstructions, which makes it difficult to rely only on the status of the up and down links.Network parameters, such as the RRH load, congestion level, link quality metric, average number of hops, and throughput, should be considered, as well, and this makes the problem formulation extremely complicated.

The approach for managing these challenges is here based on a joint optimization of the Energy and Spectral Efficiencies (ESE) in a scenario in which multi-hop D2D connections can provide diffused connectivity without a centralized support, keeping a low complexity and given the respect of constraints regarding the power, number of hops and mutual interference. The choice of a Reinforcement Learning (RL) approach, as the Q-learning, for addressing the update of a large set of network and resource allocation parameters responds to the necessity of (i) low computational complexity, (ii) possibility of a distributed implementation, and (iii) dynamic fast adaptivity for a problem with a large number of parameters. On the other hand, the selection of the ESE as the performance measure for appreciating the potential of the presented approach is clearly justified by the necessity of (i) guaranteeing the best effort in terms of throughput and, at the same time, (ii) preserving the energy in the D2D mobile devices in a critical scenario.

Taking into account all these aspects, there is a need for novel approaches that would allow a simple but effective control of the resources, minimizing the interference and optimizing the multi-hop routing, while being light-weight enough to be implemented on computationally-limited D2D devices. Therefore, here, we propose a joint ESE approach based on a light-weight partially distributed and centralized (hybrid) machine learning approach as a novel and promising way to solve the connectivity problem. This approach exploits distributed agents, i.e., OSA UEs, at a local level for executing the RL process, while the deployed command center acts as a centralized radio access network (C-RAN) for coordination and optimization of RRH performance and the RL utility parameters. In our solution, RL and, more specifically Q-Learning, is implemented as a lightweight and model-free algorithm for addressing the low complexity, adaptivity and flexibility features highly desirable in a disaster scenario, as already observed.

The main contributions of this work are summarized as follows:Joint ESE enhancement for D2D communication, including the impact of multiple hops: existing works, such as Reference [6,7,8], have investigated the trade-off between energy and spectral efficiency in the context of PSNs but without exploiting a cost function for the joint ESE with multiple hops. On the other hand, our work improves this ESE measure by applying Q-learning [9], in which the reward function reflects the link quality, i.e., interference level, power consumption, and congestion. Moreover, we apply Q-learning in a hybrid fashion (partially distributed and centralized) where the learning algorithm is applied at two types of agents: learner agents, which are the distributed devices, i.e., OSA UEs, and scheduler agents, which are the RRHs. We remark that the proposed approach is not meant to achieve the global optimal solution of the joint ESE w.r.t. all the variables because of the set of necessary simplifications and assumptions described in the paper but it is expected to provide a feasible way for approaching local maxima for the system performance.Most of the existing routing protocols in ad-hoc networks rely just on the status of the links (up or down); on the other hand, our proposed ML-based approach finds out the best path by considering the network parameters, i.e., RRH load, congestion level, link quality metric, average number of hops, and throughput. Optimizing the routing path contributes to make the agents behave so that both energy and spectral efficiency are enhanced simultaneously and dynamically. This aspect is crucial for the optimization process and it is reflected in the definition of the energy and spectral efficiency to be maximized, which incorporates the number of hops in the formulation.As a result, our approach outperforms the baseline (non-optimized) algorithm by 67% in terms of joint ESE. This translates into more UEs being alive over the number of iterations (i.e., time) as compared to the baseline algorithm. In addition, our adaptive algorithm helps to improve the energy efficiency of connected UEs by 30%. In addition, our proposed framework for a disaster scenario reduces latency by almost 50%

The paper is structured as follows: Section 2 discusses the state-of-the-art solutions, highlighting the differences with our approach. Section 3 presents the system model with the parameters and notations; Section 4 develops the problem formulation, the related constraints and the performance functions selected for our purpose. Section 5 describes in detail the approach for performing, in practice, the optimization of the cost function; Section 6 presents the numerical results. Finally, the conclusions are presented in Section 7.

## 2. Related Works

Over the last few years, work related to disaster communication management has attracted increased attention. Several systems, e.g., SafeCity and M-Urgency, are currently in use for allowing people to communicate during critical and emergency situations. SafeCity allows receiving live mobile video stream of crises and emergency situations [10]. M-Urgency allows users to use iOS or Android to stream live reports over the cellular network to a local rescue point and delivers real-time position through GPS to confirm an immediate and appropriate help for victims [11]. A stringent limitation of such systems is that they need the network infrastructure to be functional in disaster situations. However, the communication infrastructure may not be available to users in disaster areas, which makes communication difficult between victims and rescue teams.

In such a context, D2D communication can help to effectively use the radio resources for collecting useful information from different nodes in the disaster area. In earlier works, Babun et al. [6,7] developed a multi-hop based D2D communication for the public safety (PS) application. Their proposed multi-hop based communication allows for extending the coverage of the network for disaster scenarios. Simulation results show that their approach helps to increase the energy efficiency with the increasing number of hops. Moreover, they assess the trade-off between spectral and energy efficiency where with the increase of transmit power, energy efficiency decreases to achieve the same level of spectral efficiency. However, achieving joint ESE is essential but genrally not considered. Ali et al. [8] proposed a D2D communication mechanism and the use of multi-hop communications in public safety scenarios; they modified the nature of proximity service and obtained several benefits in disaster scenarios, such as low end-to-end delays, energy savings, and extension of cellular range, via UE-to-UE relaying. However, their proposed method is limited to the trade-off between energy efficiency and spectral efficiency and also the routing aspects of the system are not considered.

While joint ESE optimization has been addressed in several works concerning LTE networks (e.g., Reference [12]), to the best of our knowledge, very few papers specifically address joint ESE in D2D enabled cellular networks. Recently, [13] proposed an optimal scheme for D2D-enabled mobile-traffic offloading in a D2D overlay cellular networks with the purpose of maximizing the ESE of the network considering maximal cellular user outage and D2D transmitter power constraint. While such work provides useful insight for optimizing ESE in D2D enabled cellular networks, their focus is on the overall network performance, and not on that of the D2D users and the dissemination of critical information via multi-hop routing, as in our case.

Finally, the last few years have seen the emergence of machine learning based approaches for resource allocation and interference mitigation in D2D enabled networks (e.g., Reference [14] and the references therein). However, in the literature [15,16,17,18,19,20,21,22], we observe that light-weight, time critical on-line mechanisms for adapting resource allocation and enhancing ESE are not available. Furthermore, such works use typically centralized approaches that do not exploit the OSA resources available in this type of scenario; when the approach is partly distributed, as in Reference [23], Q-learning is exploited just for the system throughput. Then, in Reference [24], an autonomous learning method for joint channel and power level selection by D2D pairs in heterogeneous cellular networks is proposed, where D2D pairs operate on the shared cellular/D2D channels; the goal of each device pair is to select jointly the wireless channel and power level to maximize its reward, defined as the difference between the achieved throughput and the cost of power consumption, constrained by its minimum tolerable signal-to-interference-plus-noise ratio requirements. In Reference [25], the authors proposed a machine learning scheme for energy efficient resource allocation in 5G heterogeneous cloud radio access networks. Their centralized resource allocation scheme uses online learning, which guarantees interference mitigation and maximizes energy efficiency, while maintaining QoS requirements for all users. Simulation results show that the proposed resource allocation solution can mitigate interference, and increase energy and spectral efficiency significantly. However, so far, such online machine learning approaches have not been applied to optimizing the joint ESE or do not consider the multi-hop routing aspects.

In order to summarize the context in which our work is presented, the existing literature is divided in terms of energy efficiency, spectral efficiency, scenarios without full base station support, and routing for multi-hop D2D communication; therefore, our motivation comes from the lack of a coverage of all these aspects and characteristics in the same system.

## 3. System Model

The basic architecture of the D2D communication underlaying C-RAN consists of the following elements: (a) a Base band unit (BBU) pool/proximity server, (b) some RRHs, (c) Cellular users (CUs), and (d) D2D users, similar to the scenario in Reference [26].

Figure 1 shows a PS scenario where some active RRHs communicate with the C-RAN server of the deployed command center. The BBU pool/proximity server consists of BBUs that can be treated as a virtual BS (VBS) to enable the functions of the network. The fronthaul (FH) links help to connect the RRHs to the proximity server. Some UAVs, acting as RRHs, are also deployed for communication.

Due to the out-of-radio coverage situation, D2D devices need to communicate with each other in a multi-hop fashion. There are several types of interference (both inter-cell and intra-cell) in this scenario, i.e., interference from D2D users to RRHs, from cellular users to D2D users and from one D2D link to another. Our goal is to maximize the overall energy and spectral efficiency of the network exploiting multi-hop D2D communication and maintaining the communication links between CUs and active RRHs. So, interference to these links should be minimized.

We assume that there are *K* adjacent cells, {k=1,2,3,…,K}. In cell *k*, the set of D2D and CU links are denoted as $D_{k}$ and $C_{k}$, respectively. A key-point of this paper is that a link connecting two D2D users or a device to an RRH can be part of a path made of multiple links, activated by a set of D2D devices. Therefore, not all the active D2D devices are transmitting their own data, but some can operate simply as relays in the multi-hop communication path. The set of multi-hop paths for the D2D devices in the *k*-th cell is denoted as $\Lambda_{k}$, and, to each path $\lambda \in \Lambda_{k}$, it is associated the corresponding set of D2D links $L_{\lambda ,k}$; since a D2D device can belong just to one multi-hop path, we have that ${⋃}_{\lambda \in \Lambda_{k}}L_{\lambda ,k}=D_{k}$. In addition, D2D devices operating as relays can transmit and receive data on different channels, and they are supposed to reuse the channels available for standard CUs. The set of available channels in the generic cell *k* is denoted as $M_{k}$, including D2D single connections, which can be carried out with LTE-Direct or WiFi-Direct technologies.

A summary of the notations is also reported in Table 1.

## 4. Problem Formulation

In this section, we aim to achieve the formulation of the energy and spectral efficiencies with a set of constraints in order to achieve the enhancement of the system exploiting D2D, possibly multi-hop, communication, concurrent with the standard cellular CU links. In the sequel, it will be clear that the limiting factor is constituted by the mutual interference among CU and/or D2D devices. The result of the optimization, which will be the subject of the next Section 5, is supposed to provide:The assignment of the channels in $M_{k}$ for each cell *k*; the set of channels will be common and reused by all the cells.The selection of the multi-hop D2D connections, taking advantage from the energy savings and from the reuse of the cellular resources.The assignment of the powers of CU and D2D devices, taking into account the used technology and the power/rate constraints.

Let us start expressing the achievable spectral efficiency (SE) of the D2D link *d* in the *k*-th cell ($d\in D_{k},k=\left\{1,2,\cdots ,K\right\}$) on the set of available channels $M_{k}$, given by
(1)$$SE_{D2D,d,k}=\frac{1}{|M_{k}|}\sum_{m\in M_{k}}\log_{2}\left(1+\frac{\gamma_{D2D,d,k}^{m}\cdot p_{D2D,d,k}^{m}\cdot g_{d,k}^{m}}{I_{ID,d,k}^{m}+I_{ED,d,k}^{m}+I_{IC,d,k}^{m}+I_{EC,d,k}^{m}+P_{N}}\right),$$
where $|M_{k}|$ is the number of channels, $p_{D2D,d,k}^{m}$ is the transmission power of the D2D transmitter in the *d*-th link of the *k*-th cell, $\gamma_{D2D,d,k}^{m}$ is the D2D binary channel selection indicator, $P_{N}$ is the noise power, and $g_{d,k}^{m}$ is the D2D signal channel gain; the channel selection indicator $\gamma_{D2D,d,k}^{m}$ is set to 1 when the *m*-th channel is reused as the transmit channel in this link; otherwise, $\gamma_{D2D,d,k}^{m}=0$. The terms that compose the total interference are described in the following list:$I_{ID,d,k}^{m}$, the intra-cell interference on the *d*-th D2D link from the other D2D links in the *m*-th channel since D2D devices can reuse the same channel multiple times in the same cell:
(2)$$I_{ID,d,k}^{m}=\sum_{\hat{d}\in D_{k}\setminus \left\{d\right\}}\gamma_{D2D,\hat{d},k}^{m}\cdot p_{D2D,\hat{d},k}^{m}\cdot g_{\hat{d},d,k}^{m},$$
where each term in the sum is the intra-cell interference from the $\hat{d}$-th D2D interferer ($\hat{d}\neq d$).$I_{ED,d,k}^{m}$, the inter-cell interference caused by external D2D links in the *m*-th channel:
(3)$$I_{ED,d,k}^{m}=\sum_{k^{\prime }\in K\backslash \left\{k\right\}}\sum_{\hat{d}\in D_{k^{\prime }}}\gamma_{D2D,\hat{d},k^{\prime }}^{m}\cdot p_{D2D,\hat{d},k^{\prime }}^{m}\cdot g_{\hat{d},d,k^{\prime },k}^{m},$$
where each term in the double sum is the interference on the *m*-th channel from the D2D interferer in the $k^{\prime }$-th cell to the receiver of the *d*-th D2D link in the *k*-th cell ($k^{{}^{\prime }}\neq k$).$I_{IC,d,k}$, the intra-cell interference caused by CUs in the *m*-th channel:
(4)$$I_{IC,d,k}=\sum_{c\in C_{k}}\gamma_{CU,c,k}^{m}p_{CU,c,k}^{m}\cdot h_{c,d,k}^{m},$$
where $\gamma_{CU,c,k}^{m}$ is the CU binary channel selection indicator, equal to 1 if the *c*-th device in the *k*-th cell occupies channel *m* and each term in the sum is the intra-cell interference from the *c*-th CU interferer to the D2D receiver of the *d*-th link (a different symbol *h* is used for the channel gains between a CU transmitter and a D2D device). We remark that only one term in the sum will be different from 0 (so $\gamma_{CU,c,k}^{m}=1$ for one *c* and 0 for the others) since the links cannot be reused by more CU terminals in the same cell.$I_{EC,d,k}$, the inter-cell interference caused by CUs in the *m*-th channel:
(5)$$I_{EC,d,k}=\sum_{k^{\prime }\in K\backslash \left\{k\right\}}\sum_{c\in C_{k}}\gamma_{CU,c,k^{\prime }}^{m}p_{CU,c,k^{\prime }}^{m}\cdot h_{c,d,k^{\prime },k}^{m},$$
where each term in the sum denotes the inter-cell interference from the *c*-th CU interferer in the $k^{\prime }$-th cell to the *d*-th D2D link in the *k*-th cell.

Similarly, the SE of the *c*-th CU in the *k*-th cell can be expressed as
(6)$$SE_{CU,c,k}=\frac{1}{|M_{k}|}\sum_{m\in M_{k}}\log_{2}\left(1+\frac{\gamma_{CU,c,k}^{m}\cdot p_{CU,c,k}^{m}\cdot h_{c,k}^{m}}{I_{IDC,c,k}^{m}+I_{EDC,c,k}^{m}+I_{EC,c,k}^{m}+P_{N}}\right),$$
where $p_{CU,c,k}^{m}\cdot h_{c,k}^{m}$ is the signal power received by the associated RRH in the *m*-th channel. Again, the terms that compose the total interference are classified as follows:$I_{IDC,c,k}^{m}$, the intra-cell interference caused by D2D links in the same *k*-th cell, namely
(7)$$I_{IDC,c,k}^{m}=\sum_{d\in D_{k}}\gamma_{D2D,d,k}^{m}\cdot p_{D2D,d,k}^{m}\cdot {\bar{h}}_{c,d,k}^{m},$$
where each term in the sum is the intra-cell interference from the *d*-th D2D interferer on the *m*-th channel. The symbol ${\bar{h}}_{c,d,k}^{m}$ denotes the channel gain between D2D transmitter and CU link (the RRH in this case), which is different from the previous $h_{c,d,k}^{m}$, defined between the CU transmitter and the D2D receiver.$I_{EDC,c,k}^{m}$, the inter-cell interference caused by D2D pairs in the other cells,
(8)$$I_{EDC,c,k}^{m}=\sum_{k^{\prime }\in K\setminus \left\{k\right\}}\sum_{d\in D_{k^{\prime }}}\gamma_{D2D,d,k^{\prime }}^{m}\cdot p_{D2D,d,k^{\prime }}^{m}\cdot {\bar{h}}_{c,d,k,k^{\prime }}^{m},$$
where each term is the inter-cell interference on the *m*-th channel from the *d*-th D2D interferer in the $k^{\prime }$-th cell.$I_{EC,c,k}^{m}$, the inter-cell interference caused by CUs in adjacent cells, or
(9)$$I_{EC,c,k}^{m}=\sum_{k^{\prime }\in K\backslash \left\{k\right\}}\sum_{c^{\prime }\in C_{k^{\prime }}}\gamma_{CU,c^{\prime },k^{\prime }}^{m}\cdot p_{CU,c^{\prime },k^{\prime }}^{m}\cdot y_{c,c^{\prime },k,k^{\prime }}^{m},$$
where each term is the inter-cell interference from the $c^{\prime }$-th cellular interferer in the $k^{\prime }$-th cell (a different symbol *y* is used for the channel gains between a CU device and the RRH associated with another CU).

On the other hand, the Energy Efficiency (EE (bit/J)) for the single devices is computed considering the total rate in the channel bandwidth *B* divided by the consumed power, as
(10)$$EE_{D2D,d,k}=\frac{B\cdot SE_{D2D,d,k}}{p_{D2Dtot,d,k}},$$
and
(11)$$EE_{CU,c,k}=\frac{B\cdot SE_{CU,c,k}}{p_{CUtot,c,k}},$$
for the D2D and CU devices, respectively. W.r.t. the SE definitions, in which the powers are just the emitted powers; here, we have introduced the total powers, including the circuit consumption, given by
(12)$$p_{D2Dtot,d,k}=\frac{1}{\eta }p_{D2D,d,k}+2\cdot p_{cir},$$
and
(13)$$p_{CUtot,c,k}=\frac{1}{\eta }p_{CU,c,k}+p_{cir},$$
respectively. The circuit power of both the D2D transmitter and receiver is denoted by $2p_{cir}$, η is the power amplifier (PA) efficiency (0<η<1), and the circuit power of the transmitter UE is just a single term $p_{cir}$ [27].

Now, we are ready to write the SE and EE for the overall network, which are the basis for the evaluation of the method and its rationale. The overall SE ([bit/s/Hz]) in the generic *k*-th cell can be expressed as
(14)$$SE_{k}=\sum_{c\in C_{k}}\left(SE_{CU,c,k}\right)+\sum_{\lambda \in \Lambda_{k}}\left(\min_{d\in L_{\lambda ,k}}\left\{SE_{D2D,d,k}\right\}\right),$$
where the minimum of the spectral efficiencies among those associated with the links in a generic multi-hop D2D path represents the maximum achievable data throughput in the path, as the final rate has to be clearly adapted to the minimum among the theoretical capacities of the consecutive links. On the other hand, the overall energy efficiency [bit/J] is
(15)$$EE_{k}=\frac{\sum_{c\in C_{k}}B\cdot SE_{CU,c,k}+\sum_{\lambda \in \Lambda_{k}}B\min_{d\in L_{\lambda ,k}}\left\{SE_{D2D,d,k}\right\}}{\sum_{c\in C_{k}}p_{CUtot,c,k}+\sum_{d\in D_{k}}p_{D2Dtot,d,k}},$$
where $p_{CUtot,c,k}=\left(1/\eta \right)\sum_{m\in M_{k}}p_{CU,c,k}^{m}+p_{cir}$ is the total power spent by each CU, and $p_{D2Dtot,d,k}=\left(1/\eta \right)\sum_{m\in M_{k}}p_{D2D,d,k}^{m}+2p_{cir}$ is the total transmission power spent by each D2D terminal, according to Equations (Equation 13) and (Equation 12). Of course, at the second term of the denominator in Equation (Equation 15), related to the D2D devices with multi-hop paths, the sum takes into account all the powers spent by the D2D devices involved in a single path, including the relays.

Now, in order to reveal the role of the number of hops in the optimization, we propose the following approximation for Equation (Equation 15): the term $\min_{d\in L_{\lambda ,k}}\left\{SE_{D2D,d,k}\right\}$, which is challenging to manage in the optimization process, is approximated by $\left(\sum_{d\in L_{\lambda ,k}}SE_{D2D,d,k}\right)/\left|L_{\lambda ,k}\right|$, where $|L_{\lambda ,k}|$ is the number of hops in the corresponding path. This approximation is equivalent to considering all the rates in each link of a multi-hop path approximately equal, at least for their impact on the optimization function (in any case the real final rate assignment will respect the limitation given by the link with the minimum rate). This assumption has an additional impact, i.e., it mitigates the adoption of a large number of hops in the system, for their impact on the interference with other devices, and, on the SE, when we use many relays. Therefore, we express the new overall EE function for each cell *k* as
(16)$$EE_{k}^{\prime }=\frac{1}{\sum_{c\in C_{k}}p_{CU,c,k}+\sum_{d\in D_{k}}p_{D2Dtot,d,k}}\times B\left(\sum_{c\in C_{k}}SE_{CUtot,c,k}+\sum_{\lambda \in \Lambda_{k}}\sum_{d\in L_{\lambda ,k}}\frac{SE_{D2D,d,k}}{|L_{\lambda ,k}|}\right),$$
where the peculiarity is indeed the role of the factor $|L_{\lambda ,k}|$, i.e., the number of hops in each path involving D2D devices: this factor limits the adoption of the multi-hop feature to the cases in which it is really advantageous for the overall system, avoiding excessive proliferation of mutual interference phenomena and spectral efficiency losses.

In addition, we remark that D2D links (the second term in Equation (Equation 16) are reusing channels already occupied by CU terminals (contributing to ESE by means of the first term in Equation (Equation 16), and this constitutes a potential gain on the overall spectral efficiency; at the same time, the activation of links in the second term of Equation (Equation 16) increases the interference, causing a decrease of the first term. In this trade-off, the optimization process presented in Section 5 is supposed to find the correct working point.

Now, the problem for each cell *k* is given by the maximization of Equations (Equation 16) and (Equation 15), or joint ESE, with the following set of constraints {C1,C2,C3,C4,C5} regarding the maximum power, the minimum service rate and the binary channel allocation indicators:(17)$$\begin{array}{cc} \; & C1:p_{D2Dtot,d,k}\leq p_{max}\\ \; & C2:p_{CUtot,c,k}\leq p_{max}\\ \; & C3:SE_{D2D,d,k}\geq SE_{D2D,min}\\ \; & C4:SE_{CU,c,k}\geq SE_{CU,min}\\ \; & C5:\left\{\gamma_{CU,c,k}^{m},\gamma_{D2D,d,k}^{m}\right\}\in \left\{0,1\right\}, \end{array}$$
where C1 and C2 are the maximum transmission power constraints for the D2D and cellular users, respectively, C3 and C4 are the constraints for the minimum level of SE for the cellular users and D2D users, and C5 is the constraint for binary channel selection indicators.

## 5. The Proposed Method for Dynamic Adaptation of Joint Energy-Spectral Efficiency

Reinforcement learning (RL) is a powerful tool for adapting the system reaction in a dynamic environment, like D2D communications. In addition, it is characterized by low complexity, which is crucial for the use in D2D devices, and it is suited to the implementation of fully or partially distributed systems, as in our case; here, we are not considering strategies where adaptability has a heavy computational load and the set of parameters requires a fully centralized approach. One of the challenges in RL is the trade-off between exploitation and exploration. The actions are performed by agents on a trial-and-error basis during the interaction with the environment; the agents need to exploit the information collected by the learning algorithm, and they also need to explore new actions and states for finding better policies and reach learning convergence, i.e., in our case, to achieve the optimal level of ESE, which maximizes the reward in the long run. Moreover, our proposed RL approach selects the best path for multi-hop D2D routing considering important network parameters, i.e., RRH load, congestion level, link quality metric, average number of hops, and throughput. By optimizing the routing path, the proposed algorithm helps the agents to behave so that both energy and spectral efficiency are optimized. In this work, we have selected the Q-learning algorithm, as it has low computational complexity and execution time w.r.t. other variants [28], and, consequently, it is very well suited to the use in critical scenarios and D2D mobile devices.

Figure 2 shows the rationale behind the Q-learning process designed for maximizing the energy spectral efficiency. On the right side of Figure 2, the Q-learning process will operate on:the link qualities, which are determined by the interference levels, congestion levels at the RRH nodes, responsible for delays and throughput degradation, and powers;the number of hops, trying to limit them within a maximum for a positive impact on interference and power consumption.

On the left side of Figure 2, we see the two factors to be enhanced, with the arrows that highlight where an impact is expected from the factors included in the Q-learning process. The choice has been to focus on the interference and powers for their concurrent and constrasting impact on the overall SE (Equation (Equation 14)) and EE (Equation (Equation 16)). The objective is to intercept the Q-learning capability of maximizing its cost (reward) function since it is clear that increasing the powers causes a twofold impact on the SE, which tends to increase, and on the EE, which decreases; at the same time, the interference increases, with an impact on the SE. The congestion level is introduced for a more realistic evaluation of the impact on the network and we will be able to appreciate also some results concerning the latency. Finally, it is important to observe, for the relation with the formulation of the final function (Equation (Equation 16)), the role of the factor $|L_{\lambda ,k}|$ in the learning process: using more paths for the same link will be discouraged in the process because of its impact on the interference and congestion in the interested nodes without a rate increase. Therefore, it is expected that the learning process will add new hops only when the impact on the link quality will be really limited, so generating an enhancement of the spectral efficiency.

### 5.1. Network Parameters and Components in the Proposed Method

#### 5.1.1. RRH Load

Determining the RRH with the least amount of load is useful in a disaster scenario since the least loaded RRH can send feedback rapidly and can be freed for load advertising again [29]. The RRH load depends on the volume of traffic that the RRH has just processed, its current load, and its previous estimated load; this is calculated, as a function of a time step or iteration *t*, as
(18)$$L_{c}\left[t\right]=\Xi \frac{v[t]}{QL}+\left(1-\Xi \right)L_{c}\left[t-1\right],$$
where Ξ∈[0,1] is a weighting coefficient for choosing the impact of the new load measurement. The term v[t] is the volume of traffic that the RRH has just processed, and QL is the maximum queue length of the RRH.

#### 5.1.2. Congestion Level

The utilization *U* of generic link *l* is defined as follows:(19)$$U\left(l,t\right)=\frac{\sum_{i\in succ(l,t)}Size_{i}}{BW_{l}},$$
where succ(l,t) denotes the number of packets traversed link *l* successfully during time *t*, and $Size_{i}$ denotes the size of packet *i*. $BW_{l}$ is the capacity of link *l*, obtained by the spectral efficiencies derived in Section 4 and multiplied by the bandwidth *B*. Thus, we define the congestion level (CL) as
(20)CL(l,t)=βU(l,t)+(1−β)CL(l,t−1),
where the parameter β∈[0,1].

#### 5.1.3. Link Quality Metric

We define the link quality metric (LQM) for link *l* operating on a generic channel as
(21)$$LQM_{l}=\left(1-{\left(\frac{1}{2}\right)}^{n}\right)\left(I_{r_{l}}+p_{l}\right)+{\left(\frac{1}{2}\right)}^{n}CL\left(l,t\right),$$
where $I_{r_{l}}$ is the interference level, and $p_{l}$ is the total power consumption of a generic D2D or CU device, as defined in Equations (Equation 12) and (Equation 13) and used in Equation (Equation 16). The reason behind the definition of $LQM_{l}$ is our intention to capture interferences, power consumption, and congestion in computing the link quality. A larger value of *n* gives more importance to interference; on the other hand, for n=1, interference and congestion have the same weight (0.5 each). Smaller $LQM_{l}$ values for a given link reflect better quality.

Then, we define the path quality (PQ) of a path having *L* hops from a UE (D2D or CU) to an RRH as
(22)$$PQ=\sum_{l=1}^{L}\frac{{\left(LQM_{l}\right)}^{n_{H}}}{1-LQM_{l}},$$
where $n_{H}$ is the hop distance of the link starting from the RRH [29].

A low value of PQ reflects a good quality path and viceversa. The values of $LQM_{l}$ vary between 0 and 1. The ratio $\frac{{\left(LQM_{l}\right)}^{n_{H}}}{1-LQM_{l}}$ increases significantly for high values of $LQM_{l}$. Thus, PQ gives more importance/weight to links in the neighborhood of RRHs. This means that paths with high quality links in the neighborhood of RRHs will be preferred over other paths.

#### 5.1.4. Throughput

The average throughput is defined as the sum of the total amount of bits successfully received by all active users in the system divided by the product of the number of cells in the system and the transmission time interval (TTI) (which for LTE is 1 ms),
(23)$$Throughput=\frac{\sum_{c=1}^{C}\sum_{u=1}^{U}\beta_{u}^{c}}{K\times T_{sim}},$$
where *K* is the total number of cells, $T_{sim}$ is the simulation time per run, and $\beta_{u}^{c}$ is the number of bits received with success by user *u* in cell *c*. Throughput in Equation (Equation 23) should approach the maximum spectral efficiencies defined in Equations (Equation 1) and (Equation 6) multiplied by the bandwidth *B*. In the learning process, the throughput is generated considering a full queue of packets from each node in order to look at the network potential and it is maximized through the minimization of LQM. We remark that the reuse of a channel for a D2D link is considered in the process, but it is limited again by the role of interference in LQM as it should be in the allocation process.

#### 5.1.5. State

The set of states is represented by $S=\{S_{n},S_{p}\}$, where $S_{n}$ is the set of an UE’s neighboring nodes, and $S_{p}$ is the set of packets to be forwarded.

#### 5.1.6. Actions

**Forward:** Forwards a packet with selected transmit power level, $A_{f}=a(s_{j}|s_{i}),j\in I$; execution of $a(s_{j}|s_{i})$ means that UE *i* forwards a packet to UE j∈I with a given transmit power level and sends feedback to the predecessor. *I* denotes the set of the *i*’s neighboring UEs. The transmit power $p_{dc}$ is selected in the range of $\left[0,p_{max}\right]$.**Drop:** Drop, $A_{d}$, drops the data packet.

#### 5.1.7. Reward Function

We define the reward function (Rf) as follows:(24)$$Rf=\left\{\begin{array}{cc} \frac{LQM_{l}}{1-LQM_{l}}+LQM_{l}\times A & \mathrm{for}\;n\leq N,\\ \frac{LQM_{l}}{1-LQM_{l}} & n>N, \end{array}\right.$$
where *N* is the maximum number of hops in the link, and *n* is the current hop starting from the RRH. *A* assumes a predefined constant value; a large enough value of *A* (e.g., 100) allows to differentiate good paths from poor paths and eliminate the poor ones from the routing tables.

### 5.2. Proposed Reinforcement Learning for Dynamic Adaptation

One of the RL strategies for multi-agent based scheduling is based on multiple independent learners. Each agent learns independently based on local states and local rewards. This strategy may lead to an anomalous situation for action selection because there is no communication among the agents and they do not have any real view of the entire system. In this case, an agent learns according to its local information, and a coordination mechanism is required. We consider two types of agents in our environment, e.g., scheduler agent (the distributed UEs) and learner agent (the RRHs). The scheduler agents submit their local rewards to a learner agent. The learner agent collects the rewards and updates an utility table that holds the corresponding efficiency of executing action. Then, the learner agent sends the updated utility table to the scheduler agents which can then make their decisions. In each case, the data transmission energy has been considered. In the sequel, each step of the learning method is explained.

#### 5.2.1. RRH Selection Algorithm

RRHs advertise the load to the network. The first step is to find out the RRH with the minimum load. Algorithm 1 shows the steps of the RRH selection algorithm, where the input is RRH advertisements and the output is the least loaded RRH.
**Algorithm 1** RRH selection algorithm**Input**: RRHADV**Output**: Least loaded RRH**while** Battery lifetime is not equal to zero **do**    Receive RRHADV from a RRH    Update corresponding entry in RRHTable    Calculate the RRH load:$$\;\;\;\;\;\;\;\;\;\;\;\;\;\;\;\;\;\;\;\;\;\;\;\;\;\;\;\;\;\;\;\;\;\;\;\;\;\;\;\;\;\;\;\;\;\;\;\;\;\;\;\;\;\;\;L_{c}\left[t\right]=\Xi \frac{v[t]}{QL}+\left(1-\Xi \right)L_{c}\left[t-1\right]$$    BestRRH ← RRH with the minimum load**end while**

#### 5.2.2. Next-hop Selection Algorithm

D2D devices have the packet with RRH destination $RRH_{d}$. UEs need to find out the next hop to $RRH_{d}$. Now, until the battery lifetime is not equal to zero, D2D devices receive a packet with the destination RRH. Then, the agents/UEs determines the next-hop corresponding to the path with the smallest path quality and sends the packet to the neighbor, *j*. After that, the sending agents receive feedback/reward from the neighbor *j*. Finally, the Q-value for Q-learning (see Section 5.2.3) is updated. Algorithm 2 shows the steps for selecting the next hop for a given D2D device *i*.
**Algorithm 2** Next hop selection algorithm at D2D device *i***Input**: Packet with RRH destination $RRH_{d}$**Output**: Best next hop to $RRH_{d}$**Variables**: RoutingTable, j**while** Battery lifetime is not equal to zero **do**    Receive a packet with destination RRH    Determine the next-hop corresponding to the path with the smallest path quality (PQ):$$\;PQ=\sum_{l=1}^{L}\frac{{\left(LQM_{l}\right)}^{n_{H}}}{1-LQM_{l}}$$    Send packet to j with selected level of $p_{dc}$    Receive feedback/reward, Rf from j$$\;Rf=\left\{\begin{array}{cc} \frac{LQM_{l}}{1-LQM_{l}}+LQM_{l}\times A & \mathrm{for}\;n\leq N.\\ \frac{LQM_{l}}{1-LQM_{l}} & n>N. \end{array}\right.$$    Update the *Q* value for Q-learning    Update the corresponding entry in the table, RRHTable**end while**

Figure 3 depicts the multi-hop routing scenario considering Algorithms 1 and 2. The figure illustrates the RRHADV (RRH advertisements) in the network where UEs receive RRHADV and select the least loaded RRH, as well as the hop selection, based on the path quality (PQ), where UEs select the next hop corresponding to the path with the smallest PQ. For example, in the figure, there are two paths with PQ=0.25 and PQ=0.50; UEs select the hop with the smallest one (PQ=0.25) according to the proposed algorithm.

#### 5.2.3. Q-Learning for Dynamic Adaptation

We apply Q-learning for optimizing the routing path and ESE. In our case, the components are:*Agent:* Each UE denotes an agent responsible for executing the online learning algorithm.*Environment:* The application represents the environment in our approach. Interaction between the agents and the environment is achieved by executing actions and receiving a reward function.*Action:* Agent action is the currently executed application task on the UEs.*State:* A state describes a particular scenario of the environment based on some application oriented variables.*Policy:* Agent policy determines what action will be selected in a particular state. This policy determines which action to execute at the perceived state and focuses more on exploration or exploitation, depending on the selected setting of the learning algorithm.*Value function:* This function defines what is good for an agent over the long run. It is built upon the reward function values over time, and its quality exclusively depends on the reward function.*Reward function:* This function provides a mapping of the agent’s state and the corresponding action with a reward value that contributes to the performance.

In the *Q* learning framework, the agents learn the utility of performing various actions over time steps using the local information [30]. Here, the main network parameters and components, e.g., RRH load, congestion level, link quality metric, throughput, and overall energy-spectral efficiency, are dynamically changing. In this dynamic environment, agents need to learn the right action to perform at particular states. As shown in Algorithm 3, here, the UEs try to maximize the utility and maintain a *Q* matrix for the value functions. Initially, all the *Q* values of the matrix are zeros. At each state, the UEs perform an action; depending on the reward function and the *Q* value, they move to the next state which maximizes the system performance.
**Algorithm 3** Proposed reinforcement learning algorithmInitialize Q(s,a)=0 where *s* is the set of states and *a* is the set of actions**while** Battery lifetime is not equal to zero **do**    Determine current state    Select action *a* based on policy$$\;\frac{e^{Q(s,a)}/\omega }{\sum_{a}\left(e^{Q(s,a)/\omega }\right)}$$    Execute the selected action    Calculate the reward$$\;Rf=\left\{\begin{array}{cc} \frac{LQM_{l}}{1-LQM_{l}}+LQM_{l}\times A & \mathrm{for}\;n\leq N.\\ \frac{LQM_{l}}{1-LQM_{l}} & n>N. \end{array}\right.$$    Calculate the learning rate$$\;\varphi =\frac{Z}{visited(s,a)}$$    Calculate *Q* value for the executed action$$\;Q_{t+1}\left(s_{t},a_{t}\right)=\left(1-\varphi \right)Q_{t}\left(s_{t},a_{t}\right)+\varphi \left(Rf\left(s_{t+1}\right)+\Gamma V_{t}\left(s_{t+1}\right)\right)$$    Calculate the value function for the executed action$$\;V_{t+1}\left(s_{t}\right)=\max_{a\in A}Q_{t+1}\left(s_{t},a\right)$$    Update the utility table of the scheduler agent$$\;U\left(q\right)=\left(1-\Upsilon \right)U\left(q\right)+\Upsilon \sum_{i}Rf_{i}$$    Move to the next state based on the executed action**end while**

The *Q* value is updated as follows for the (state, action) pair:(25)$$Q_{t+1}\left(s_{t},a_{t}\right)=\left(1-\varphi \right)Q_{t}\left(s_{t},a_{t}\right)+\varphi \left(r_{t+1}\left(s_{t+1}\right)+\Gamma V_{t}\left(s_{t+1}\right)\right),$$
(26)$$V_{t+1}\left(s_{t}\right)=\max_{a\in A}Q_{t+1}\left(s_{t},a\right),$$
where $Q_{t+1}\left(s_{t},a_{t}\right)$ is the update of the *Q* value at time t+1 after executing action *a* at time step *t*. $r_{t+1}$ is the immediate reward after executing the action *a* at time *t*, $V_{t}$ is the value function for node at time *t*, and $V_{t+1}$ is the value function at time t+1. The term $\max_{a\in A}Q_{t+1}\left(s_{t},a\right)$ is the maximum *Q* value after performing an action from the action set *A* for the agent *i*. The parameter Γ is the discount-factor which can be set to a value in [0,1]; for higher Γ values, the agent relies more on the future than the immediate reward. Finally, φ is the learning rate parameter which can be set to a value in [0,1] [31]; it controls the rate at which an agent tries to learn by giving more or less weight to the previously learned utility value. When φ is close to 1, the agent gives more priority to the previously learned utility value.

Here, we use soft-max strategy and Boltzmann distributions for the exploration and exploitation [32]. The probability of selecting an action *a* in state *s* is proportional to $e^{Q(s,a)}\omega$. That is, at state *s*, agents select an action based on the probability
(27)$$\frac{e^{Q(s,a)}/\omega }{\sum_{a}\left(e^{Q(s,a)/\omega }\right)},$$
where ω is the temperature constant. If ω>0, the agents will focus on choosing the actions randomly, i.e., exploration. On the other hand, if ω→0, the best action based on Q-values is chosen, i.e., exploitation. The learning rate, φ, is slowly decreased in order to take into account the impact of the visited state-action pair, i.e.,
(28)$$\varphi =\frac{Z}{visited(s,a)},$$
where *Z* is the positive constant, and visited(s,a) is the number of visited state-action pairs so far.

#### 5.2.4. Updating the Utility Table

In each time step, the learner agent receives the reward from all scheduler agents and updates the utility table, U(q):(29)$$U\left(q\right)=\left(1-\Upsilon \right)U\left(q\right)+\Upsilon \sum_{i}Rf_{i},$$
where Υ is learning factor, and $Rf_{i}$ is the reward vector generated by the *i*-th agent [33].

After updating the utility table, the learner agent sends it back to the scheduler agents; they will generate the rewards and submit the reward vector to the learner agent. Table 2 shows the main parameters of the reinforcement learning.

## 6. Performance Evaluation

In this section, we present the simulation results in order to discuss the performance of our proposed approach in terms of joint ESE and remaining energy of the network.

We evaluate our approach against existing variants of RL, i.e., state-action-reward-state-action (SARSA), Q(λ), and SARSA(λ) [34], and observe that we outperform them all. SARSA shows an increase of EE until a similar number of stages as with our approach, but then EE decreases very rapidly due to the number of depleted UEs. Indeed, in Q-learning, the Q-value is updated with the maximum valued action at the next state; on the other hand, in SARSA, the update is dependent upon the action that is actually taken at the next state. While SARSA is suitable when the agents focus on exploration [35], it is risky for dynamic scenarios like ours, where there exist a number of depleted UEs over time, as well as interferences. The Q(λ) algorithm is similar to Q-learning except for the eligibility traces. Q(λ) stops learning at the iteration where the agent selects the exploratory action and eligibility traces for all state-action pairs are set to zero, which is also not suitable for dynamic scenarios like ours [36].

### 6.1. Simulation Setup

Rouil et al. [37] provide an implementation of LTE D2D functionalities (direct communication, direct discovery, and out of coverage D2D synchronization) for NS3. We extend their implemented model by adding C-RAN functionalities (BBUs and RRH/UAVs), for communication purposes and an ad-hoc routing protocol for multi-hop D2D communication.

The channel gain between the transmitter *i* and the receiver *j* is proportional to $d_{i,j}^{-2}{\left|h_{i,j}\right|}^{2}$, where $d_{i,j}$ is the distance between the transmitter *i* and the receiver *j*. $h_{i,j}$ is the complex Gaussian channel coefficient that satisfies $h_{i,j}∼CN\left(0,1\right)$ [38]. Each simulation starts with the UEs of random initial amounts of battery charge between 1 and 800 (mAh). The simulation results are averaged over 10 simulation runs. The location of the UEs are generated randomly in each simulation run.

Table 3 shows the simulation parameters used in the simulation.

### 6.2. Results for Stand-Alone EE and SE

In this subsection, stand-alone denotes the optimization of either EE or SE.

#### 6.2.1. Stand-Alone EE Performance Evaluation

Figure 4 shows the stand-alone EE optimization, calculated by Equation (Equation 15) and non-optimized SE, calculated by Equation (Equation 14). We can observe that our proposed method helps to increase EE until 160 iterations. Then, with the increasing number of depleted UEs, EE progressively decreases.

The Q(λ) algorithm is similar to Q-learning except for the eligibility traces; Q(λ) stops learning at the iteration where the agent selects the exploratory action and eligibility traces for all state-action pairs are set to zero, which is also not suitable for dynamic scenarios like ours [36]. So, the algorithm Q(λ) shows lower EE compared to our approach and SARSA up to 210 iterations, after which it is slightly better than SARSA (Q(λ) takes a little time to pass SARSA due to the accumulating traces of non-greedy actions).

SARSA(λ) is similar to SARSA with the eligibility traces, as well; with SARSA(λ), EE is lower compared to the other methods, although a similar trend can be observed for both SARSA(λ) and Q(λ). However, SARSA(λ) takes over SARSA after 205 iterations due to the accumulating eligibility traces.

Furthermore, we can observe that, from the 50-th iteration till to the 250-th, all learning algorithms behave similarly, until reaching the convergence level. After reaching the convergence learning level, our proposed reinforcement learning outperforms the other methods.

Finally, since SE is not optimized, it rapidly decreases after 50 iterations due to the depleted number of UEs and interferers, and this motivates the need for joint ESE optimization, as shown in Section 6.3.

#### 6.2.2. Stand-Alone SE Performance Evaluation

Figure 5 shows the stand-alone SE optimization and unoptimized EE. We can observe that the optimized SE remains maximized for a longer time as compared to the baseline technique. However, with the increasing number of depleted UEs, the SE decreases (after approximately 170 iterations). But EE without optimization goes down even more rapidly. Our proposed learning method outperforms the three other learning variants. SARSA performs better than Q(λ) and SARSA(λ) until 270 iterations; then, SE decreases rapidly due to SARSA’s focus on exploration. Q(λ) performs better than SARSA(λ) at every iteration. Our approach benefits from the exploration-exploitation strategy and heuristic learning rate update mechanism, which helps the agents to behave in an adaptive manner in the environment for achieving higher SE. On the other hand, the baseline (unoptimized) EE decreases rapidly after only 75 iterations as it does not have any dynamic adaptation to face the changes in the environment.

### 6.3. Performance Evaluation of Joint ESE

Figure 6a shows the joint ESE using our proposed approach over the number of iterations. It yields the best results among all evaluated approaches, achieving convergence at iteration 138 and remaining at the same level until iteration 165. Then, joint ESE decreases because of the energy depletion in the UEs. At this point, we can sacrifice delay and focus on joint ESE to keep the network alive as long as possible. Here, we observe the same trend as in all methods for stand-alone EE and stand-alone SE. Again, our approach performs better than the others, thanks to the algorithm’s exploration-exploitation strategy and heuristic learning rate. Initially, SARSA performs better than Q(λ) and SARSA(λ) but is overtaken by Q(λ) at approximately 260 iterations and by SARSA(λ) at approximately 300 iterations, thanks to their accumulating traces.

On the other hand, for the baseline (unoptimized) method, the joint ESE decreases very sharply after only 50 iterations. Our proposed method outperforms the joint ESE by 67% as compared to the baseline algorithm.

Moreover, our proposed approach outperforms all others in terms of the percentage of alive UEs. Figure 6b shows that for the joint ESE, optimized by means of our proposed approach, 98% of UEs are alive after 125 iterations, whereas only 79% are alive at the same number of iterations without optimization.

Finally, Figure 7 shows the cumulative distribution function (CDF) for the energy efficiency of the connected UEs. We adopt the Monte-Carlo method to calculate the CDF to obtain the numerical results. Here, we observe that our proposed method for energy efficiency outperforms the baseline one up to 30%.

### 6.4. Delay and Execution Time

Figure 8 shows the end-to-end delay of the network as a function of the number of hops. Here, end-to-end delay is estimated as the time between the availability of a packet at the transmitter and at the receiver in ms. We observe that, when increasing the number of hops, the end-to-end delay increases linearly in both frameworks, i.e., D2D using C-RAN and D2D without C-RAN. We compare our framework with the D2D without C-RAN framework, and we observe that the end-to-end delay decreases in our proposed framework. The reason for this improvement is that the data traffic passes through BBUs that have higher processing power and can deliver the data to the UEs more rapidly, while D2D communication without C-RAN functionality is characterized by higher latency. Our proposed framework yields 50% reduced latency compared to the traditional infrastructure. This result highlights the contribution of our proposed framework in delay-sensitive disaster scenario, e.g., a zone subject to a terrorist attack.

Finally, Table 4 shows the execution times for the different methods: we can observe that our proposed method outperforms the others in terms of execution time. We use the hardware configuration of Intel(R) Core (TM) i7-4790 3.60 GHz CPU, 8 GB RAM and in a 64-bit Windows 10 operating system. In addition, Simulator::Schedule is used in NS-3 for calculating the execution time. The execution time is the average of five complete simulation runs consists of 400 iterations each.

## 7. Conclusions

In a disaster scenario where the network coverage is not fully ensured, our proposed method for D2D communication underlaying C-RAN helps to provide improved joint ESE as compared to the baseline approach. Simulation results show that our optimized network outperforms the baseline (unoptimized) one by almost 67% in terms of joint energy-spectral efficiency, and our algorithm helps to achieve approximately 30% energy efficiency for connected UEs compared to the baseline technique. Moreover, using C-RAN infrastructure, our adaptive algorithm helps to reduce the latency by 50%. We also evaluated our proposed methods against other variants of reinforcement learning and showed that our proposed method outperforms them in terms of both stand-alone and joint ESE.

In the future, we will explore other adaptive optimization methods, i.e., mixed integer optimization with adaptive partition and multi-directional search in the network. Our intention with such methods is to optimize the parameters, i.e., energy efficiency, spectral efficiency, latency, outage probability, etc., in a way that some parameters can be relaxed in order to focus on the most critical parameters depending on the requirements of the environment.

## Figures and Tables

**Figure 1 sensors-20-06692-f001:**
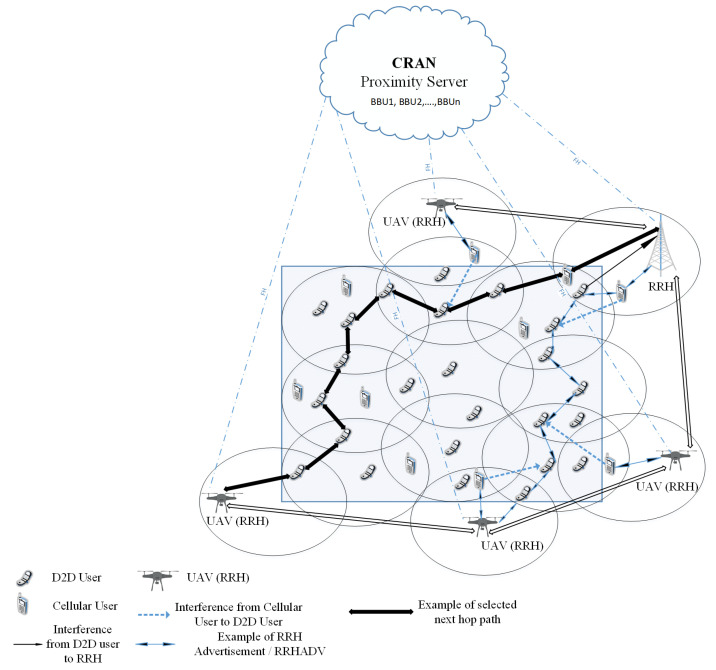
The overall scenario of multi-hop Device-to-Device (D2D) communications underlaid centralized radio access network (C-RAN) for public safety application.

**Figure 2 sensors-20-06692-f002:**
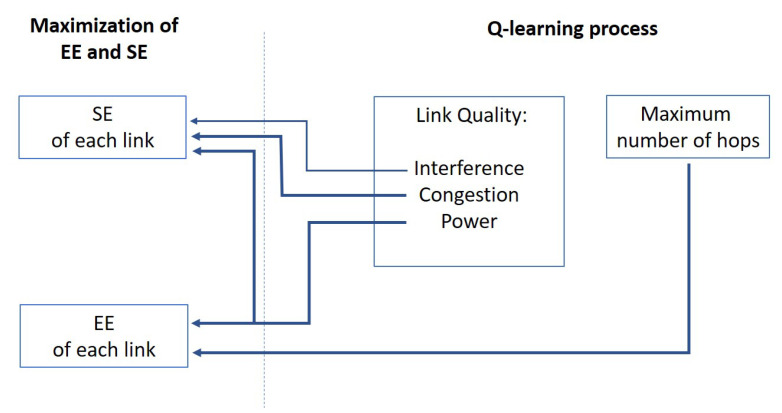
Rationale of the Q-learning process for the maximization of energy and spectral efficiencies (ESE).

**Figure 3 sensors-20-06692-f003:**
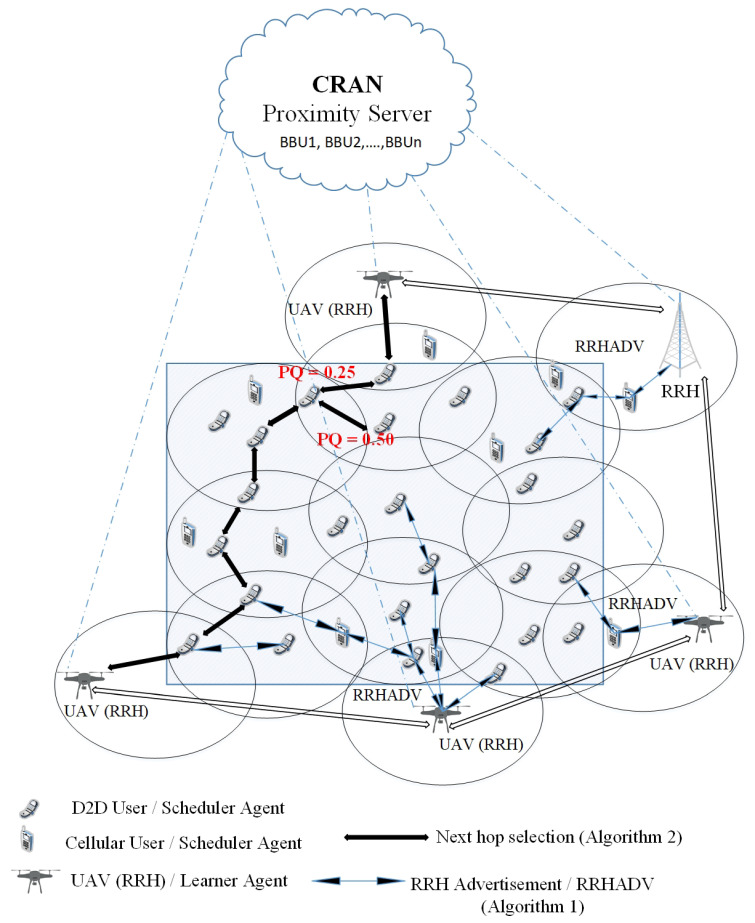
Illustration for Algorithm 1 and Algorithm 2. Algorithm 1: Remote radio head (RRH) advertisement (RRHADV) from RRHs. Scheduler agents/user equipment (UEs) receive RRHADV from RRHs to select the least loaded RRH/BestRRH. Algorithm 2: Next-hop selection for UEs/Scheduler agents and routing toward the destination RRH. UEs select the next-hop to the path with the smallest value of path quality (PQ).

**Figure 4 sensors-20-06692-f004:**
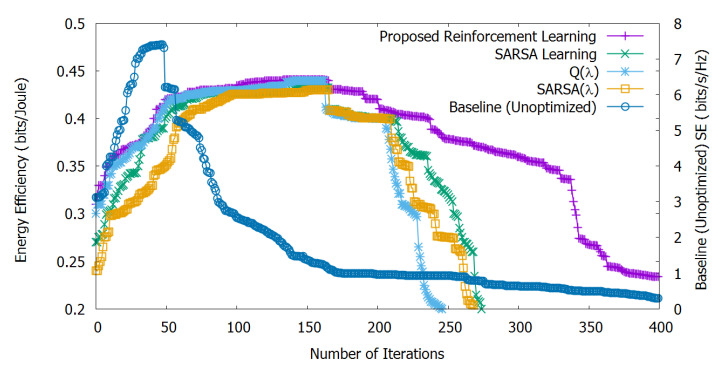
Stand-alone energy efficiency performance evaluation and non-optimized spectral efficiency. The proposed method helps to increase Energy Efficiency (EE) until 160 iterations; afterwards, energy efficiency progressively decreases due to the increasing number of depleted UEs.

**Figure 5 sensors-20-06692-f005:**
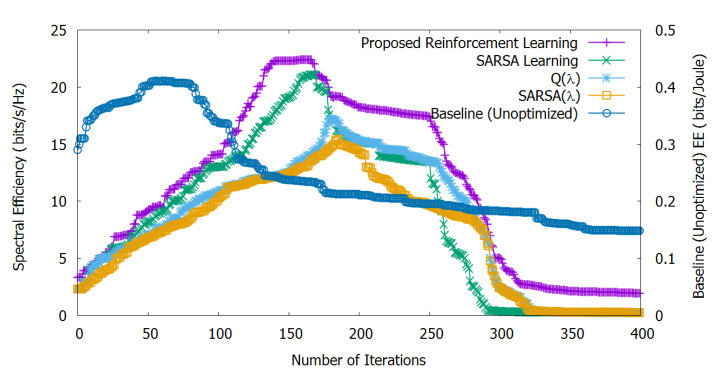
Stand-alone spectral efficiency performance evaluation and unoptimized energy efficiency. Spectral efficiency optimized by means of the proposed method remains maximized for a longer time as compared to the baseline technique; it decreases after approximately 170 iterations due to the increasing number of depleted UEs.

**Figure 6 sensors-20-06692-f006:**
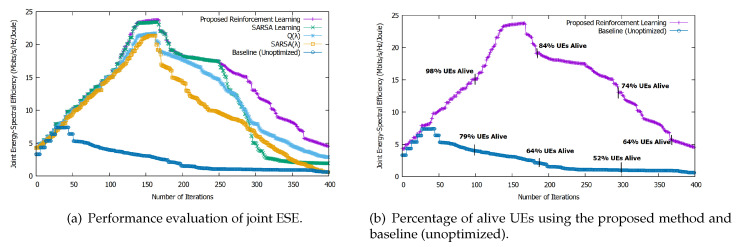
Performance evaluation of joint ESE and percentage of alive UEs. (**a**) Performance evaluation of joint ESE. (**b**) Percentage of alive UEs using the proposed method and baseline (unoptimized).

**Figure 7 sensors-20-06692-f007:**
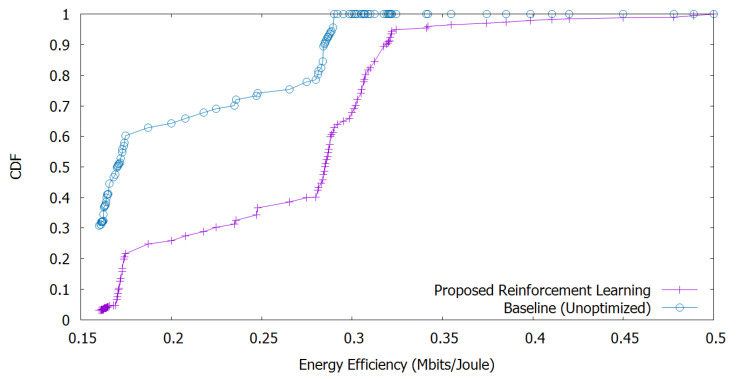
Cumulative distribution function (CDF) of energy efficiency of the connected UEs.

**Figure 8 sensors-20-06692-f008:**
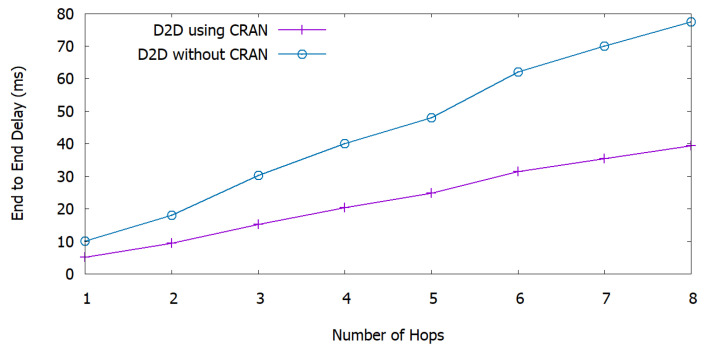
End-to-end delay of the network over number of hops.

**Table 1 sensors-20-06692-t001:** Notations for the system model.

*K*	Number of cells
$C_{k}$	Set of CU devices in cell *k*
$D_{k}$	Set of D2D devices in cell *k*
$M_{k}$	Set of available channels in cell *k*
$\Lambda_{k}$	Set of multi-hop paths for the D2D devices in cell *k*
$L_{\lambda ,k}$	Set of D2D links in the λ-th path in $\Lambda_{k}$
*c*	Index for a generic CU belonging to $C_{k}$
*d*	Index for a generic D2D belonging to $D_{k}$ or $L_{\lambda ,k}$
λ	Index of a given multi-hop path in $\Lambda_{k}$
|·|	Number of elements in the sets $C_{k}$, $D_{k}$, $M_{k}$, $\Lambda_{k}$

**Table 2 sensors-20-06692-t002:** Reinforcement learning parameters.

Parameter	Value
$Q_{t+1}\left(s_{t},a_{t}\right)$	update of the *Q* value at time t+1
$\max_{a\in A}Q_{t+1}\left(s_{t},a\right)$	Maximum *Q* value
$r_{t+1}$	Immediate Reward
$v_{t}$	Value function
Γ	Discount factor
φ	Learning rate
ω	Temperature constant

**Table 3 sensors-20-06692-t003:** Simulation parameters and their values.

Parameter	Symbol	Value
Area		1000 m × 1000 m
Total Number of UEs		60
Cell radius		300 m
Intercell distance		500 m
Number of RRHs		5
Maximum Tx power	$p_{max}$	23 dBm
Bandwidth	*B*	180 kHz
Constant circuit power	$p_{cir}$	10 dBm
Thermal noise power	$P_{N}$	$10^{-7}$ W
Battery capacity		800 mAh
Congestion level par.	β	0.8
Reward Function par.	*A*	100
Learning rate	φ	0.5
Learning rate par.	*Z*	1
Learning factor	Υ	0.5
Discount factor	Γ	0.5
Minimum SE for D2D	$SE_{D2D,min}$	1.90
Minimum SE for CU	$SE_{CU,min}$	0.30
Max number of hops	*N*	2

**Table 4 sensors-20-06692-t004:** Execution times for the methods.

Method	Execution Time (s)
Proposed method	2.13
Q(λ)	2.50
SARSA	2.85
SARSA(λ)	2.90

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
