# Peer review of "Q-Learning Based Joint Energy-Spectral Efficiency Optimization in Multi-Hop Device-to-Device Communication"

_sensors, 2020, doi:10.3390/s20226692_

Round 1
Reviewer 1 Report
This paper addresses Q-learning based approach to energy-spectral efficiency optimization in Multi-Hop d2d communication. It looks natural to apply to RL for complicated optimization problem for a valid solution. Also, the results look valid in that they have some advantage over other compared solutions. However, the proposed algorithm indeed does not solve the objective function that is presented in the paper but is changed for convenience of obtaining a solution. Thus, the resulting solution does not necessarily guarantee to solve the optimal solution for the claimed objective function. It is strongly suggested to discuss the optimality of the obtained results with respect to the objective function presented in the formulation.
Author Response
Comments:
This paper addresses Q-learning based approach to energy-spectral efficiency optimization in Multi-Hop d2d communication. It looks natural to apply to RL for complicated optimization problem for a valid solution. Also, the results look valid in that they have some advantage over other compared solutions. However, the proposed algorithm indeed does not solve the objective function that is presented in the paper but is changed for convenience of obtaining a solution. Thus, the resulting solution does not necessarily guarantee to solve the optimal solution for the claimed objective function. It is strongly suggested to discuss the optimality of the obtained results with respect to the objective function presented in the formulation.
Reply: thank you for your comments. We agree that the proposed algorithm does not solve directly the objective function that is presented in Sect. 4 and it does not guarantee the global optimality of the solution; however, the Q-learning based approach implements a formulation adapted for (i) addressing the computational complexity of the problem and, in particular, (ii) making feasible a partially distributed, more realistic implementation of the allocation strategy exploiting measures that are available at the nodes of the networks. In fact this is one of the key-points and motivations behind the paper, i.e. implementing an allocation strategy in an emergency context in which the connectivity might be totally or partially damaged. On one hand, the optimal solution of the cost function would lead to an NP-hard problem with the necessity of performing a set of simplifications; on the other hand, a solution involving the entire set of parameters would typically require a fully centralized approach. In the revised paper, we have clarified this aspect, which was expressed in a misleading way, leaving in Sect. 4 only the performance measures of the system (spectral and energy efficiencies to be optimized) and improving the method explanation at the beginning of Sect. 5, devoted to the ML approach.
Reviewer 2 Report
This paper proposes a Q-learning based joint energy-spectral efficiency optimization approach for D2D communication. Overall, the paper is well-written, the use of Q-learning is able to achieve good performance on energy efficiency and latency. However, the explanations of some important parts can be further improved. Some suggestions are listed below.
More explanation on how to formulate ESE function (14) should be given: what this means, why it should be formulated in this way, and how this is related to the previous formulas.
The explanation of Figure 2 can be improved. It is not easy to understand the explanation in page 9, with all the definitions given in page 11-12. It is not suggested to refer to the contents that have not mentioned in the paper yet. So it is better to give the definitions first. For example, first give a basic idea of the process without referring to any definition, then give the definitions, and finally give a final summary of the process, as what you had in page 9.
The algorithms (1-3) are not shown correctly.
What value of A is used for the reward function (22)?
As shown Figure 7 on the CDF, the proposed RL approach even has lower energy efficiency than the baseline. However, in the text, the author says the proposed approach outperforms the baseline by up to 30%. Also, in the conclusion, “our algorithm helps to achieve approx. 30% energy efficiency for connected UEs compared to the baseline technique”: this sentence is wrong.
Author Response
Reviewer-2
This paper proposes a Q-learning based joint energy-spectral efficiency optimization approach for D2D communication. Overall, the paper is well-written, the use of Q-learning is able to achieve good performance on energy efficiency and latency. However, the explanations of some important parts can be further improved. Some suggestions are listed below.
Comments:
1. More explanation on how to formulate ESE function (14) should be given: what this means, why it should be formulated in this way, and how this is related to the previous formulas.
Reply: Thank you for pointing out this weakness since it is a crucial point. We have revised Section 4, presenting the spectral and energy efficiency definitions for single devices and for the overall network since these measures are used for evaluating the impact of the Q-learning based approach. The old (14), which is formally an energy efficiency, is now presented as an approximated way for computing the overall energy efficiency, useful for understanding the rationale behind the Q-learning approach. These modifications interest the final part of Sect. 4 and the initial part of Sect. 5.
2. The explanation of Figure 2 can be improved. It is not easy to understand the explanation in page 9, with all the definitions given in page 11-12. It is not suggested to refer to the contents that have not mentioned in the paper yet. So it is better to give the definitions first. For example, first give a basic idea of the process without referring to any definition, then give the definitions, and finally give a final summary of the process, as what you had in page 9.
Reply: Thank you for this comment. We have changed this part following your suggestions and we think that this has improved the clarity of the text.
3. The algorithms (1-3) are not shown correctly.
Reply: The Algorithms (1-3) are now properly visible.
4. What value of A is used for the reward function (22)?
Reply: The value of A is shown in Table 3 and explained in Sect. 5.1.7.
5. As shown Figure 7 on the CDF, the proposed RL approach even has lower energy efficiency than the baseline. However, in the text, the author says the proposed approach outperforms the baseline by up to 30%. Also, in the conclusion, “our algorithm helps to achieve approx. 30% energy efficiency for connected UEs compared to the baseline technique”: this sentence is wrong.
Reply: Thank you for this comment. We have found that there was a plotting mistake with the data files, where we have corrected the Figure now and the proposed method outperforms approx. 30% in terms of energy efficiency for connected UEs compared to the baseline technique.
Reviewer 3 Report
This work proposes a Q-learning-based approach for routing in ad hoc networks to optimize energy and spectral efficiency simultaneously for multi-hop device to device communication.
Overall, the paper is not easy to follow due to the massive use of mathematic symbols (fomulration), abbreviations, and lack fo explanation to figures, equations, and experimental settings. Details see below:
1. The introduction can be better sorted out. The distinct challenges to be addressed by this work should be clearly stated before summarizing the main contributions.
2. There are too many acronyms/abbreviations in the manuscript, making it hard to follow.
3. The problem definition can be refined. A clearer case example might be given, with the source node and destination node of paths annotated. Some questions remain unanswered, e.g., why are the equations (esp., equation 15) defined that way in Section 4? do those formulations differ from the problem definitions in the related work.
4. The use of reinforcement learning lacks motivation. Why reinforcement learning (or more specifically, Q-learning) is necessary or suitable for the specific problem?
5. Figure 2 appears too simplistic. The meaning of arrows is unclear.
6. Among the claimed contributions, #2 and #4 largely overlap. Authors should reduce redundancies (e.g., repeated statements) in illustration.
7. The paper needs thorough proofreading. There are "Sect. ??" and redundant ")" in the manuscript.
8. Algorithms need better formating. Lines mixed up in the algorithm description.
9. Q-learning is used for dynamic adaptation. So what are the changing parts taken into account? What types of decisions can be made by each type of agents?
10. The baseline selection should be justified. For example, why [20] is not compared? Simulation details (e.g., the meaning of parameters in Table 3) can be extended.
Author Response
Reviewer-3
This work proposes a Q-learning-based approach for routing in ad hoc networks to optimize energy and spectral efficiency simultaneously for multi-hop device to device communication.
Overall, the paper is not easy to follow due to the massive use of mathematic symbols (formulation), abbreviations, and lack of explanation to figures, equations, and experimental settings. Details see below:
Comments:
1.The introduction can be better sorted out. The distinct challenges to be addressed by this work should be clearly stated before summarizing the main contributions.
Reply: Thanks for pointing out this issue. The Introduction section has been revised and partially reorganized in order to highlight the challenges and the related contributions (Line number 50-89, 96-102 and 114-115).
2.There are too many acronyms/abbreviations in the manuscript, making it hard to follow.
Reply: It is true that the paper has a large number of acronyms and symbols. We have simplified some parts avoiding the definitions not strictly necessary, mainly in Sect. 6 (Table 3).
3. The problem definition can be refined. A clearer case example might be given, with the source node and destination node of paths annotated. Some questions remain unanswered, e.g., why are the equations (esp., equation 15) defined that way in Section 4? do those formulations differ from the problem definitions in the related work.
Reply: Thank you for pointing out this issue. We have revised Section 4, presenting the spectral and energy efficiency definitions for single devices and for the overall network since these measures are used for evaluating the impact of the optimization Q-learning based approach. These SE and EE definitions are shared also by the literature in the field, even if with possible slight differences especially in the definition of the powers for the EE. The old (14), which is formally an energy efficiency, is now presented as an approximated way (this is an original contribution of the paper) for computing the overall energy efficiency, useful for understanding the rationale behind the Q-learning approach and these modifications interest the final part of Sect. 4 and the initial one of Sect. 5.
4. The use of reinforcement learning lacks motivation. Why reinforcement learning (or more specifically, Q-learning) is necessary or suitable for the specific problem?
Reply: Reinforcement learning is not the unique possible response to this problem but we consider it an appropriate candidate for three main reasons: (i) it is an approach for a realistic implementation of an optimization problem whose solution would be unfeasible with traditional strategies, (ii) it is not problem-specific and with limited complexity, (iii) it is adaptive, so suitable for a dynamic system, even fast. We have remarked on these aspects in the paper as well.
5. Figure 2 appears too simplistic. The meaning of arrows is unclear.
Reply: Fig. 2 has been changed and it reports, in a simplified and schematic way, the relations between the proposed Q-learning method, the performance measures (energy and spectral efficiency), and the main parameters in this work. We have improved its description at the beginning of Sect. 5.
6. Among the claimed contributions, #2 and #4 largely overlap. Authors should reduce redundancies (e.g., repeated statements) in illustration.
Reply: Thank you for this comment. We have removed the redundancies.
7. The paper needs thorough proofreading. There are "Sect. ??" and redundant ")" in the manuscript.
Reply: We have performed proofreading and worked on the typos.
8. Algorithms need better formating. Lines mixed up in the algorithm description.
Reply: We have worked on this, now the algorithms should be reported correctly.
9. Q-learning is used for dynamic adaptation. So what are the changing parts taken into account? What types of decisions can be made by each type of agent?
Reply: We have clarified these aspects in Sect. .5.2.3
10. The baseline selection should be justified. For example, why [20] is not compared? Simulation details (e.g., the meaning of parameters in Table 3) can be extended.
Reply: Thank you for your comment. The baseline is the non-optimized system and it is the main reference for evaluating performance enhancement. We have not found works that are easily comparable in the same public safety scenario, in particular with the exploitation of multiple hops. Finally, we have revised also Table 3, eliminating the parameters not strictly necessary and adding a brief description.
Reviewer 4 Report
Proposed manuscript deals with a very interesting topic of device to device communication management. Manuscript has a logical structure and it is clearly written but I am not sure why did the authors choose MDPI Sensors. There are some much more suitable journals in the MDPI family (Applied Sciences).
Manuscript describes an actual problem of multi-hop device to device communication, optimisation and critical management. Results are proved by validation.
I have following comments:
1) with all respect to MDPI Sensors, why did you choose this journal?
2) Figs 1, 4 and others: better charts are needed.
3) Why did you choose reinforcement learning for the optimisation? Is it really the best tool? Did you compare it to other solutions?
4) I miss some important references from last years
Proposed manuscript is clearly written and it has a logical structure. If this topic is suitable for MDPI Sensors (I am not sure about that) I recommend MINOR REVISION.
Author Response
Proposed manuscript deals with a very interesting topic of device to device communication management. Manuscript has a logical structure and it is clearly written but I am not sure why did the authors choose MDPI Sensors. There are some much more suitable journals in the MDPI family (Applied Sciences).
Manuscript describes an actual problem of multi-hop device to device communication, optimisation and critical management. Results are proved by validation.
Comments:
1.With all respect to MDPI Sensors, why did you choose this journal?
Reply: We have selected a special issue of MDPI Sensors that matches well this paper. The name of the special issue is: "Internet of Things for Smart Community Solutions", where one of their topics of interest is IoT and machine learning. Our paper rightly falls into that category.
2. Figs 1, 4 and others: better charts are needed.
Reply: Thank you for this observation. We have tried to improve the quality of the figures.
3. Why did you choose reinforcement learning for the optimisation? Is it really the best tool? Did you compare it to other solutions?
Reply: Thank you for this comment. reinforcement learning is a useful tool to do a sub-optimal, adaptive improvement of the system with parameters that may change dynamically. In our scenario, this capability is a crucial factor and, in addition, it is characterized by low complexity and it is not problem-specific. In order to maintain the same advantages and, at the same time, the same level of complexity, we have compared different variants of reinforcement learning.
4. I miss some important references from last years
Reply: We have checked the bibliography for finding similar works w.r.t. the considered scenario in the last 2 years and added to the reference.
Here are those:
Park, H.; Lim, Y. Reinforcement Learning for Energy Optimization with 5G Communications in Vehicular Social Networks.Sensors2020,20, 2361
Najla, M.; Gesbert, D.; Becvar, Z.; Mach, P. Machine Learning for Power Control in D2D Communication571Based on Cellular Channel Gains. 2019 IEEE Globecom Workshops (GC Wkshps). IEEE, 2019, pp. 1–6.
Round 2
Reviewer 3 Report
I am basically satisfied with this version.
It seems all my concerns have been addressed.